# MINI-BATCH $k$-MEANS TERMINATES WITHIN $O(d/\epsilon)$ ITERATIONS

**Gregory Schwartzman**
Japan Advanced Institute of Science and Technology (JAIST)
greg@jaist.ac.jp

## ABSTRACT

We answer the question: "Does *local* progress (on batches) imply *global* progress (on the entire dataset) for mini-batch $k$-means?". Specifically, we consider mini-batch $k$-means which terminates only when the improvement in the quality of the clustering on the sampled batch is below some threshold.

Although at first glance it appears that this algorithm might execute forever, we answer the above question in the affirmative and show that if the batch is of size $\tilde{\Omega}((d/\epsilon)^2)$, it must terminate within $O(d/\epsilon)$ iterations with high probability, where $d$ is the dimension of the input, and $\epsilon$ is a threshold parameter for termination. This is true *regardless* of how the centers are initialized. When the algorithm is initialized with the $k$-means++ initialization scheme, it achieves an approximation ratio of $O(\log k)$ (the same as the full-batch version).

Finally, we show the applicability of our results to the mini-batch $k$-means algorithm implemented in the scikit-learn (sklearn) python library.

## 1 INTRODUCTION

The mini-batch $k$-means algorithm (Sculley, 2010) is one of the most popular clustering algorithms used in practice (Pedregosa et al., 2011). However, due to its stochastic nature, it appears that if we do not explicitly bound the number of iterations of the algorithm, then it might never terminate. We show that, when the batch size is sufficiently large, using only an "early-stopping" condition, which terminates the algorithm when the local progress observed on a batch is below some threshold, we can guarantee a bound on the number of iterations that the algorithm performs which is *independent* of input size.

**Problem statement** We consider the following optimization problem. We are given an input (dataset), $X = \{x_i\}_{i=1}^n \subseteq [0,1]^d$, of size $n$ of $d$-dimensional real vectors and a parameter $k$. Note that the assumption that $X \subseteq [0,1]^d$ is standard in the literature (Arthur et al., 2011), and is meant to simplify notation (otherwise we would have to introduce a new parameter for the diameter of $X$). Our goal is to find a set $\mathcal{C}$ of $k$ centers (vectors in $[0,1]^d$) such that the following goal function is minimized:

$$\frac{1}{n} \sum_{x \in X} \min_{c \in \mathcal{C}} \|c - x\|^2$$

Usually, the $1/n$ factor does not appear as it does not affect the optimization goal, however, in our case, it will be useful to define it as such.

**Lloyd's algorithm** The most popular method to solve the above problem is Lloyd's algorithm (often referred to as the $k$-means algorithm) (Lloyd, 1982). It works by randomly initializing a set of $k$ centers and performing the following two steps: (1) Assign every point in $X$ to the center closest to it. (2) Update every center to be the mean of the points assigned to it. The algorithm terminates when no point is reassigned to a new center. This algorithm is extremely fast in practice but has a worst-case exponential running time (Arthur & Vassilvitskii, 2006; Vattani, 2011).

**Mini-batch $k$-means**  To update the centers, Lloyd's algorithm must go over the entire input at every iteration. This can be computationally expensive when the input data is extremely large. To tackle this, the mini-batch $k$-means method was introduced by Sculley (2010). It is similar to Lloyd's algorithm except that steps (1) and (2) are performed on a batch of $b$ elements sampled uniformly at random with repetitions, and in step (2) the centers are updated slightly differently. Specifically, every center is updated to be the weighted average of its current value and the mean of the points (in the batch) assigned to it. The parameter by which we weigh these values is called the *learning rate*, and its value differs between centers and iterations. In the original paper by Sculley, there is no stopping condition similar to that of Lloyd's algorithm, instead, the algorithm is simply executed for $t$ iterations, where $t$ is an input parameter.

In practice (for example in sklearn (Pedregosa et al., 2011)), together with an upper bound on the number of iterations to perform there are several "early stopping" conditions. We may terminate the algorithm when the change in the locations of the centers is sufficiently small or when the change in the goal function for several consecutive batches does not improve. We note that in both theory (Tang & Monteleoni, 2017; Sculley, 2010) and practice (Pedregosa et al., 2011) the learning rate goes to 0 over time. That is, over time the movement of centers becomes smaller and smaller, which guarantees termination for most reasonable early-stopping conditions at the limit.

Our results are the first to show extremely fast termination guarantees for mini-batch $k$-means with early stopping conditions. Surprisingly, we need not require the learning rate to go to 0.

**Related work**  Mini-batch $k$-means was first introduced by Sculley (2010) as a natural generalization to online $k$-means (Bottou & Bengio, 1994) (here the batch is of size 1). We are aware only of a single paper that analyzes the convergence rate of mini-batch $k$-means (Tang & Monteleoni, 2017). It is claimed in (Tang & Monteleoni, 2017) that under mild assumptions the algorithm has $O(1/t)$ convergence rate. That is, after $t$ iterations it holds that the current value of the goal function is within an additive $O(1/t)$ factor from the value of the goal function in some local optimum of Lloyd's algorithm. However, their asymptotic notation subsumes factors that depend on the size of the input. Taking this into account, we get a convergence rate of $\Omega(n^2/t)$, which implies, at best, a quadratic bound on the execution time of the algorithm. This is due to setting the learning rate at iteration $t$ to $O(1/(n^2 + t))$. Our results do not guarantee convergence to any local-minima, however, they guarantee an exponentially faster runtime bound.

**Our results**  We analyze the mini-batch $k$-means algorithm described above (Sculley, 2010), where the algorithm terminates only when the improvement in the quality of the clustering for the sampled batch is less than some threshold parameter $\epsilon$. That is, we terminate if for some batch the difference in the quality of the clustering before the update and after the update is less than $\epsilon$. Our stopping condition is slightly different than what is used in practice. In sklearn termination is determined based on the changes in cluster centers. In Section 5 we prove that this condition also fits within our framework.

Our main goal is to answer the following theoretical question: "Does local progress (on batches) imply global progress (on the entire dataset) for mini-batch $k$-means, even when the learning rate does not go to 0?". Intuitively, it is clear that the answer depends on the batch size used by the algorithm. If the batch is the entire dataset the claim is trivial and results in a termination guarantee of $O(d/\epsilon)$ iterations[1]. We show that when the batch size exceeds a certain threshold, indeed local progress implies global progress and we achieve the same asymptotic bound on the number of iterations as when the batch is the entire dataset. We present several results:

We start with a warm-up in Section 3, showing that when $b = \tilde{\Omega}(kd^3\epsilon^{-2})$ we can guarantee termination within $O(d/\epsilon)$ iterations[2] w.h.p (with high probability)[3]. We require the additional assumption that every real number in the system can be represented using $O(1)$ bits (e.g., 64-bit floats). The above bound holds *regardless* of how cluster centers are initialized or updated. That is, this bound holds for any center update rule, and not only for the "standard" center update rule described above. Our proof uses elementary tools and is presented to set the stage for our main result.

---

[1]This holds because the maximum value of the goal function is $d$ (Lemma 2.1).

[2]Throughout this paper the tilde notation hides logarithmic factors in $n, k, d, \epsilon$.

[3]This is usually taken to be $1 - 1/n^p$ for some constant $p \geq 1$. For our case, it holds that $p = 1$, however, this can be amplified arbitrarily by increasing the batch size by a multiplicative constant factor.

In Section 4 we show that using the standard update rule, we can achieve the same termination time with a much smaller batch size. Specifically, a batch size of $\Omega((d/\epsilon)^2 \log(nkd/\epsilon)) = \tilde{\Omega}((d/\epsilon)^2)$ is sufficient to guarantee termination within $O(d/\epsilon)$ iterations. This holds regardless of how centers are initialized and does not require any assumption on the number of bits required to represent real numbers. Our proof makes use of the fact that the standard update rule adds additional stability to the stochastic process when the learning rate is sufficiently small (but need not go to 0). Finally, in Section 5, we show that our main result also holds for the early stopping condition used in sklearn (with our learning rate). However, this results in a larger batch size and slower termination. Specifically if $b = \tilde{\Omega}((d/\epsilon)^3 k)$ we terminate within $O((d/\epsilon)^{1.5}\sqrt{k})$ iterations w.h.p.

Note that for the batch size to be reasonable, we must require that $b \leq n$, which implies that $(d/\epsilon)^2 \log(nkd/\epsilon) = O(n)$. Thus, our results only hold for a certain range of values for $k, d, \epsilon$. This is reasonable, as in practice it is often the case that $\epsilon = O(1), d \ll n$ and the dependence on the rest of the parameters is logarithmic.

**Solution quality** Applying the $k$-means++ initialization scheme to our results we achieve the same approximation ratio, $O(\log k)$ in expectation, as the full-batch algorithm. The approximation guarantee of $k$-means++ is guaranteed already in the initialization phase (Theorem 3.1 in Arthur & Vassilvitskii (2007)), and the execution of Lloyd's algorithm following initialization can only improve the solution. We show that w.h.p the global goal function is decreasing throughout our execution which implies that the approximation guarantee remains the same.

## 2 PRELIMINARIES

Throughout this paper we work with ordered tuples rather than sets, denoted as $Y = (y_i)_{i \in [\ell]}$, where $[\ell] = \{1, \ldots, \ell\}$. To reference the $i$-th element we either write $y_i$ or $Y[i]$. It will be useful to use set notations for tuples such as $x \in Y \iff \exists i \in [\ell], x = y_i$ and $Y \subseteq Z \iff \forall i \in [\ell], y_i \in Z$. When summing we often write $\sum_{x \in Y} g(x)$ which is equivalent to $\sum_{i=1}^{\ell} g(Y[i])$.

We borrow the following notation from (Kanungo et al., 2004). For every $x, y \in \mathbb{R}^d$ let $\Delta(x, y) = \|x - y\|^2$. For every finite tuple $S \subseteq \mathbb{R}^d$ and a vector $x \in \mathbb{R}^d$ let $\Delta(S, x) = \sum_{y \in S} \Delta(y, x)$.

$k$**-means** We are given an input $X = (x_i)_{i=1}^n \subseteq [0, 1]^d$ and a parameter $k$. Our goal is to find a tuple $\mathcal{C} \subseteq \mathbb{R}^d$ of $k$ centers such that the following goal function is minimized:

$$\frac{1}{n} \sum_{x \in X} \min_{C \in \mathcal{C}} \Delta(x, C)$$

Let us define for every $x \in X$ the function $f_x : \mathbb{R}^{k \cdot d} \to \mathbb{R}$ where $f_x(\mathcal{C}) = \min_{C \in \mathcal{C}} \Delta(x, C)$. We can treat $\mathbb{R}^{k \cdot d}$ as the set of $k$-tuples of $d$-dimensional vectors. We also define the following function for every tuple $A = (a_i)_{i=1}^{\ell} \subseteq X$:

$$f_A(\mathcal{C}) = \frac{1}{\ell} \sum_{i=1}^{\ell} f_{a_i}(\mathcal{C})$$

Note that $f_X$ is our original goal function. We state the following useful lemma:

**Lemma 2.1.** *For any tuple of $k$ centers $\mathcal{C} \subset [0, 1]^d$ it holds that $f_X(\mathcal{C}) \leq d$.*

*Proof.* Because $X, \mathcal{C} \subset [0, 1]^d$ it holds that $\forall x \in X, f_x(\mathcal{C}) \leq \max_{C \in \mathcal{C}} \Delta(x, C) \leq d$. Therefore $f_X(\mathcal{C}) = \frac{1}{n} \sum_{x \in X} f_x(\mathcal{C}) \leq \frac{1}{n} \cdot nd = d$.

$\square$

We state the following well known theorems:

**Theorem 2.2** (Hoeffding (1963)). *Let $Y_1, ..., Y_m$ be independent random variables such that $\forall 1 \leq i \leq m, E[Y_i] = \mu$ and $Y_i \in [a_{min}, a_{max}]$. Then*

$$Pr\left(\left|\frac{1}{m} \sum_{i=1}^{m} Y_k - \mu\right| \geq \delta\right) \leq 2e^{-2m\delta^2/(a_{max} - a_{min})^2}$$

**Theorem 2.3** (Jensen (1906)). *Let $\phi$ be a convex function, $y_1, \ldots, y_n$ numbers in its domain and weights $a_1, \ldots, a_n \in \mathbb{R}^+$. It holds that:*

$$\phi \left( \frac{\sum_{i=1}^n a_i y_i}{\sum_{i=1}^n a_i} \right) \leq \frac{\sum_{i=1}^n a_i \phi(y_i)}{\sum_{i=1}^n a_i}$$

## 3  WARM-UP: A SIMPLE BOUND

Let us first show a simple convergence guarantee which makes no assumptions about how the centers are updated. This will set the stage for our main result in Section 4, where we consider the standard update rule used in mini-batch $k$-means (Sculley, 2010; Pedregosa et al., 2011).

**Algorithm**  We analyze a generic variant of the mini-batch $k$-means algorithm, presented in Algorithm 1. Note that it a very broad class of algorithms (including the widely used algorithm of Sculley (2010)). The only assumptions we make are:

1. The centers remain within $[0,1]^d$ (the convex hull bounding $X$).

2. Batches are sampled uniformly at random from $X$ with repetitions.

3. The algorithm terminates when updating the centers does not significantly improve the quality of the solution for the sampled batch.

Items (1) and (2) are standard both in theory and practice (Sculley, 2010; Pedregosa et al., 2011; Tang & Monteleoni, 2017). Item (3) is usually referred to as an "early-stopping" condition. Early stopping conditions are widely used in practice (for example in sklearn (Pedregosa et al., 2011)), together with a bound on the number of iterations. However, our early-stopping condition is slightly different than the one used in practice. We discuss this difference in Section 5.

At first glance, guaranteeing termination for *any* possible way of updating the centers might seem strange. However, if the update procedure is degenerate, it will make no progress, at which point the algorithm terminates.

---

**Algorithm 1:** Generic mini-batch $k$-means

1 $\mathcal{C}_1 \subseteq [0,1]^d$ is an initial tuple of centers
2 **for** $i = 1$ *to* $\infty$ **do**
3 $\quad$ Sample $b$ elements, $B_i = (y_1, \ldots, y_b)$, uniformly at random from $X$ (with repetitions)
4 $\quad$ Update $\mathcal{C}_{i+1}$ (such that $\mathcal{C}_{i+1} \subseteq [0,1]^d$)
5 $\quad$ **if** $f_{B_i}(\mathcal{C}_i) - f_{B_i}(\mathcal{C}_{i+1}) < \epsilon$ **then** Return $\mathcal{C}_{i+1}$

---

**Termination guarantees for Algorithm 1**  To bound the number of iterations of such a generic algorithm we require the following assumption: every real number in our system can be represented using $q = O(1)$ bits. This implies that every set of $k$ centers can be represented using $qkd$ bits. This means that the total number of possible solutions is bounded by $2^{qkd}$. This will allow us to show that when the batch is sufficiently large, the sampled batch acts as a *sparsifier* for the entire dataset. Specifically, it means that for *any* tuple of $k$ centers, $\mathcal{C}$, it holds that $|f_{B_i}(\mathcal{C}) - f_X(\mathcal{C})| < \epsilon/4$. This implies that, for a sufficiently large batch size, simply sampling a single batch and executing Lloyd's algorithm on the batch will be sufficient, and executing mini-batch $k$-means is unnecessary. Nevertheless, this serves as a good starting point to showcase our general approach and to highlight the challenges we overcome in Section 4 in order to reduce the required batch size without compromising the running time.

Let us assume that the algorithm executes for at least $t$ iterations. That is, the termination condition does not hold for the first $t$ iterations. Our goal is to upper bound $t$.

**Parameter range**  Let us first define the range of parameter values for which the results for this section hold. Recall that $n$ is the size of the input, $k$ is the number of centers, $d$ is the dimension, $\epsilon$ is

the termination threshold. For the rest of this section assume that $b = \Omega((d/\epsilon)^2(kd + \log(nt)))$. Later we show that $t = O(d/\epsilon)$, which will imply that $b = \tilde{\Omega}(kd^3\epsilon^{-2})$ is sufficient for our termination guarantees to hold.

We state the following useful lemma which guarantees that $f_B(\mathcal{C})$ is not too far from $f_X(\mathcal{C})$ when the batch size is sufficiently large and $\mathcal{C}$ is fixed (i.e., independent of the choice of $B_i$).

**Lemma 3.1.** *Let $B$ be a tuple of $b$ elements chosen uniformly at random from $X$ with repetitions. For any fixed tuple of $k$ centers, $\mathcal{C} \subseteq [0, 1]^d$, it holds that: $Pr[|f_B(\mathcal{C}) - f_X(\mathcal{C})| \geq \delta] \leq 2e^{-2b\delta^2/d^2}$.*

*Proof.* Let us write $B = (y_1, \ldots, y_b)$, where $y_i$ is a random element selected uniformly at random from $X$ with repetitions. For every such $y_i$ define the random variable $Z_i = f_{y_i}(\mathcal{C})$. These new random variables are IID for any fixed $\mathcal{C}$. It also holds that $\forall i \in [b], E[Z_i] = \frac{1}{n}\sum_{x \in X} f_x(\mathcal{C}) = f_X(\mathcal{C})$ and that $f_B(\mathcal{C}) = \frac{1}{b}\sum_{x \in B} f_x(\mathcal{C}) = \frac{1}{b}\sum_{i=1}^{b} Z_i$.

Applying a Hoeffding bound (Theorem 2.2) with parameters $m = b, \mu = f_X(\mathcal{C}), a_{max} - a_{min} \leq d$ we get that: $Pr[|f_B(\mathcal{C}) - f_X(\mathcal{C})| \geq \delta] \leq 2e^{-2b\delta^2/d^2}$. $\square$

Using the above we can show that every $B_i$ is a sparsifier for $X$.

**Lemma 3.2.** *It holds w.h.p that for every $i \in [t]$ and for every set of $k$ centers, $\mathcal{C} \subset [0, 1]^d$, that $|f_{B_i}(\mathcal{C}) - f_X(\mathcal{C})| < \epsilon/4$.*

*Proof.* Using Lemma 3.1, setting $\delta = \epsilon/4$ and using the fact that $b = \Omega((d/\epsilon)^2(kd + \log(nt)))$, we get: $Pr[|f_B(\mathcal{C}) - f_X(\mathcal{C})| \geq \delta] \leq 2e^{-2b\delta^2/d^2} = 2^{-\Theta(b\delta^2/d^2)} = 2^{-\Omega(kd + \log(nt))}$.

Taking a union bound over all $t$ iterations and all $2^{qkd}$ configurations of centers, we get that the probability is bounded by $2^{-\Omega(kd + \log(nt))} \cdot 2^{qkd} \cdot t = O(1/n)$, for an appropriate constant in the asymptotic notation for $b$. $\square$

The lemma below guarantees *global progress* for the algorithm.

**Lemma 3.3.** *It holds w.h.p that $\forall i \in [t], f_X(\mathcal{C}_i) - f_X(\mathcal{C}_{i+1}) \geq \epsilon/2$.*

*Proof.* Let us write (the notation $\pm x$ means that we add and subtract $x$):
$$f_X(\mathcal{C}_i) - f_X(\mathcal{C}_{i+1}) = f_X(\mathcal{C}_i) \pm f_{B_i}(\mathcal{C}_i) \pm f_{B_i}(\mathcal{C}_{i+1}) - f_X(\mathcal{C}_{i+1}) \geq \epsilon/2$$
Due to Lemma 3.2 it holds that w.h.p $f_X(\mathcal{C}_i) - f_{B_i}(\mathcal{C}_i) > -\epsilon/4$ and $f_{B_i}(\mathcal{C}_{i+1}) - f_X(\mathcal{C}_{i+1}) > -\epsilon/4$. Finally due to the termination condition it holds that $f_{B_i}(\mathcal{C}_i) - f_{B_i}(\mathcal{C}_{i+1}) \geq \epsilon$. This completes the proof. $\square$

As $f_X$ is upper bounded by $d$, it holds that we must terminate within $O(d/\epsilon)$ iterations w.h.p when $b = \Omega(kd^3\epsilon^{-2}\log(nd/\epsilon))$. We state our main theorem for this Section.

**Theorem 3.4.** *For $b = \tilde{\Omega}(kd^3\epsilon^{-2})$, Algorithm 1 terminates within $O(d/\epsilon)$ iterations w.h.p.*

**Towards a smaller batch size** Note that the batch size used in this section is about a $kd$ factor larger than what we require in Section 4. This factor is required for the union bound over all possible sets of $k$ centers in Lemma 3.2. However, when actually applying Lemma 3.2, we only apply it for two centers in iteration $i$, setting $B = B_i$ and $\mathcal{C} = \mathcal{C}_i, \mathcal{C}_{i+1}$. A more direct approach would be to apply Lemma 3.1 only for $\mathcal{C}_i, \mathcal{C}_{i+1}$, which would get rid of the extra $kd$ factor. This will work when $\mathcal{C} = \mathcal{C}_i$ as $B_i$ is sampled *after* $\mathcal{C}_i$ is determined, but will fail for $\mathcal{C} = \mathcal{C}_{i+1}$ because $\mathcal{C}_{i+1}$ may depend on $B_i$. In the following section, we show how to use the fact that the learning rate is sufficiently small in order to overcome this challenge.

## 4 MAIN RESULTS

In this section, we show that we can get a much better dependence on the batch size when using the standard center update rule. Specifically, we show that a batch of size $\tilde{\Omega}((d/\epsilon)^2)$ is sufficient to guarantee termination within $O(d/\epsilon)$ iterations. We also do not require any assumption about the number of bits required to represent a real number.

**Section preliminaries** Let us define for any finite tuple $S \subset \mathbb{R}^d$ the center of mass of the tuple as $cm(S) = \frac{1}{|S|} \sum_{x \in S} x$. For any tuple $S \subset \mathbb{R}^d$ and some tuple of cluster centers $\mathcal{C} = (\mathcal{C}^\ell)_{\ell \in [k]}$ it implies a *partition* $(S^\ell)_{\ell \in [k]}$ of the points in $S$. Specifically, every $S^\ell$ contains the points in $S$ closest to $\mathcal{C}^\ell$ and every point in $S$ belongs to a single $\mathcal{C}^\ell$ (ties are broken arbitrarily). We state the following useful observation:

**Observation 4.1.** *Fix some $A \subseteq X$. Let $\mathcal{C}$ be a tuple of $k$ centers, $S = (S^\ell)_{\ell \in [k]}$ be the partition of $A$ induced by $\mathcal{C}$ and $\overline{S} = (\overline{S}^\ell)_{\ell \in [k]}$ be any other partition of $A$. It holds that $\sum_{j=1}^k \Delta(S^j, \mathcal{C}^j) \leq \sum_{j=1}^k \Delta(\overline{S}^j, \mathcal{C}^j)$.*

Let $\mathcal{C}_i^j$ denote the location of the $j$-th center in the beginning of the $i$-th iteration. Let $(B_i^\ell)_{\ell \in [k]}$ be the partition of $B_i$ induced by $\mathcal{C}_i$ and let $(X_i^\ell)_{\ell \in [k]}$ be the partition of $X$ induced by $\mathcal{C}_i$.

We analyze Algorithm 1 when clusters are updated as follows: $\mathcal{C}_{i+1}^j = (1 - \alpha_i^j)\mathcal{C}_i^j + \alpha_i^j cm(B_i^j)$, where $\alpha_i^j$ is the *learning rate*. Note that $B_i^j$ may be empty in which case $cm(B_i^j)$ is undefined, however, the learning rate is chosen such that $\alpha_i^j = 0$ in this case ($\mathcal{C}_{i+1}^j = \mathcal{C}_i^j$). Note that the learning rate may take on different values for different centers, and may change between iterations. In the standard mini-batch $k$-means algorithm (Sculley, 2010; Pedregosa et al., 2011) the learning rate goes to 0 over time. This guarantees termination for most reasonable stopping conditions.

As before, we assume that the algorithm executes for at least $t$ iterations and upper bound $t$. We show that the learning rate need not go to 0 to guarantee termination when the batch size is sufficiently large. Specifically, we set $\alpha_i^j = \sqrt{b_i^j/b}$, where $b_i^j = \left| B_i^j \right|$, and we require that $b = \Omega((d/\epsilon)^2 \log(ndtk))$.

**Proof outline** In our proof, we use the fact that a sufficiently small learning rate enhances the stability of the algorithm, which in turn allows us to use a much smaller batch size compared to Section 3. Let us define the auxiliary value $\overline{\mathcal{C}}_{i+1}^j = (1 - \alpha_i^j)\mathcal{C}_i^j + \alpha_i^j cm(X_i^j)$. This is the $j$-th center at step $i + 1$ if we were to use the entire dataset for the update, rather than just a batch. Note that this is only used in the analysis and not in the algorithm.

Recall that in the previous section we required a large batch size because we could not apply Lemma 3.1 when $B = B_i$ and $\mathcal{C} = \mathcal{C}_{i+1}$ because $\mathcal{C}_{i+1}$ may depend on $B_i$. To overcome this challenge we use $\overline{\mathcal{C}}_{i+1}$ instead of $\mathcal{C}_{i+1}$. Note that $\overline{\mathcal{C}}_{i+1}$ only depends on $\mathcal{C}_i, X$ and is independent of $B_i$ (i.e., we can fix its value before sampling $B_i$). We show that for our choice of learning rate it holds that $\overline{\mathcal{C}}_{i+1}, \mathcal{C}_{i+1}$ are sufficiently close, which implies that $f_X(\mathcal{C}_{i+1}), f_X(\overline{\mathcal{C}}_{i+1})$ and $f_{B_i}(\mathcal{C}_{i+1}), f_{B_i}(\overline{\mathcal{C}}_{i+1})$ are also sufficiently close. This allows us to use a similar proof to that of Lemma 3.3 where $\overline{\mathcal{C}}_{i+1}$ acts as a proxy for $\mathcal{C}_{i+1}$. We formalize this intuition in what follows.

First, we state the following useful lemmas:

**Lemma 4.2** (Kanungo et al. (2004)). *For any set $S \subseteq \mathbb{R}^d$ and any $C \in \mathbb{R}^d$ it holds that $\Delta(S, C) = \Delta(S, cm(S)) + |S|\,\Delta(C, cm(S))$.*

**Lemma 4.3.** *For any $S \subseteq X$ and $C, C' \in [0,1]^d$, it holds that: $|\Delta(S, C') - \Delta(S, C)| \leq 2\sqrt{d}\,|S|\,\|C - C'\|$.*

*Proof.* Using Lemma 4.2 we get that $\Delta(S, C) = \Delta(S, cm(S)) + |S|\,\Delta(cm(S), C)$ and that $\Delta(S, C') = \Delta(S, cm(S)) + |S|\,\Delta(cm(S), C')$. Thus, it holds that $|\Delta(S, C') - \Delta(S, C)| = |S| \cdot |\Delta(cm(S), C') - \Delta(cm(S), C)|$. Observe that for two vectors $x, y \in \mathbb{R}^d$ it holds that $\Delta(x, y) = (x - y) \cdot (x - y)$. Let us switch to vector notation and bound $|\Delta(cm(S), C') - \Delta(cm(S), C)|$.

$$
\begin{aligned}
&|\Delta(cm(S), C') - \Delta(cm(S), C)| \\
&= |(cm(S) - C') \cdot (cm(S) - C') - (cm(S) - C) \cdot (cm(S) - C)| \\
&= |-2cm(S) \cdot C' + C' \cdot C' + 2cm(S) \cdot C - C \cdot C| \\
&= |2cm(S) \cdot (C - C') + (C' - C) \cdot (C' + C)| \\
&= |(C - C') \cdot (2cm(S) - (C' + C))| \\
&\leq \|C - C'\|\|2cm(S) - (C' + C)\| \leq 2\sqrt{d}\|C - C'\|
\end{aligned}
$$

Where in the last transition we used the Cauchy-Schwartz inequality. $\qquad\square$

First, we show that due to our choice of learning rate $\mathcal{C}^j_{i+1}, \overline{\mathcal{C}}^j_{i+1}$ are sufficiently close.

**Lemma 4.4.** *For it holds w.h.p that* $\forall i \in [t], j \in [k], \|\mathcal{C}^j_{i+1} - \overline{\mathcal{C}}^j_{i+1}\| \leq \frac{\epsilon}{10\sqrt{d}}$.

*Proof.* Note that $\mathcal{C}^j_{i+1} - \overline{\mathcal{C}}^j_{i+1} = \alpha^j_i(cm(B^j_i) - cm(X^j_i))$. Let us fix some iteration $i$ and center $j$. To simplify notation, let us denote: $X' = X^j_i, B' = B^j_i, b' = b^j_i, \alpha' = \alpha^j_i$. Although $b'$ is a random variable, in what follows we treat it as a fixed value (essentially conditioning on its value). As what follows holds for *all* values of $b'$ it also holds without conditioning due to the law of total probabilities.

For the rest of the proof, we assume $b' > 0$ (if $b' = 0$ the claim holds trivially). Let us denote by $\{Y_\ell\}_{\ell=1}^{b'}$ the sampled points in $B'$. Note that a randomly sampled element from $X$ is in $B'$ if and only if it is in $X'$. As batch elements are sampled uniformly at random with repetitions from $X$, conditioning on the fact that an element is in $B'$ means that it is distributed uniformly over $X'$. Thus, it holds that $\forall \ell, E[Y_\ell] = \frac{1}{|X'|} \sum_{x \in X'} x = cm(X')$ and $E[cm(B')] = \frac{1}{b'} \sum_{\ell=1}^{b'} E[Y_\ell] = cm(X')$. Our goal is to bound $Pr[\|cm(B') - cm(X')\| \geq \frac{\epsilon}{10\alpha'\sqrt{d}}]$, we note that it is sufficient to bound the deviation of every coordinate by $\epsilon/(10\alpha'd)$, as that will guarantee that:

$$\|cm(B') - cm(X')\| = \sqrt{\sum_{\ell=1}^{d}(cm(B')[\ell] - cm(X')[\ell])^2} \leq \sqrt{\sum_{\ell=1}^{d}(\frac{\epsilon}{10\alpha'd})^2} = \frac{\epsilon}{10\alpha'\sqrt{d}}$$

We note that for a single coordinate, $\ell$, we can apply a Hoeffding bound with parameters $\mu = cm(X')[\ell], a_{max} - a_{min} \leq 1$ and get that:

$$Pr[|cm(B')[\ell] - cm(X')[\ell]| \geq \frac{\epsilon}{10\alpha'd}] \leq 2 \cdot e^{-\frac{2b'\epsilon^2}{100(\alpha')^2d^2}}$$

Taking a union bound we get that

$$Pr[\|cm(B') - cm(X')\| \geq \frac{\epsilon}{10\alpha'\sqrt{d}}]$$

$$\leq Pr[\exists \ell, |cm(B')[\ell] - cm(X')[\ell]| \geq \frac{\epsilon}{10\alpha'd}] \leq 2d \cdot e^{-\frac{2b'\epsilon^2}{100(\alpha')^2d^2}}$$

Using the fact that $\alpha' = \sqrt{b'/b}$ together with the fact that $b = \Omega((d/\epsilon)^2 \log(ntkd))$ (for an appropriate constant) we get that the above is $O(1/ntk)$. Finally, taking a union bound over all $t$ iterations and all $k$ centers per iteration completes the proof. $\qquad\square$

Let us now use the above lemma to bound the goal function when cluster centers are close.

**Lemma 4.5.** *Fix some $A \subseteq X$. It holds w.h.p that* $\forall i \in [t], |f_A(\overline{\mathcal{C}}_{i+1}) - f_A(\mathcal{C}_{i+1})| \leq \epsilon/5$

*Proof.* Let $S = (S^\ell)_{\ell \in [k]}, \overline{S} = (\overline{S}^\ell)_{\ell \in [k]}$ be the partitions induced by $\mathcal{C}_{i+1}, \overline{\mathcal{C}}_{i+1}$ on $A$. Let us expand the expression:

$$f_A(\overline{\mathcal{C}}_{i+1}) - f_A(\mathcal{C}_{i+1}) = \frac{1}{|A|} \sum_{j=1}^{k} \Delta(\overline{S}^j, \overline{\mathcal{C}}^j_{i+1}) - \Delta(S^j, \mathcal{C}^j_{i+1})$$

$$\leq \frac{1}{|A|} \sum_{j=1}^{k} \Delta(S^j, \overline{\mathcal{C}}^j_{i+1}) - \Delta(S^j, \mathcal{C}^j_{i+1})$$

$$\leq \frac{1}{|A|} \sum_{j=1}^{k} 2\sqrt{d} |S^j| \|\overline{\mathcal{C}}^j_{i+1} - \mathcal{C}^j_{i+1}\| \leq \frac{1}{|A|} \sum_{j=1}^{k} |S^j| \epsilon/5 = \epsilon/5$$

Where the first inequality is due to Observation 4.1, the second is due Lemma 4.3 and finally we use Lemma 4.4 together with the fact that $\sum_{j=1}^{k} |S^j| = |A|$. Using the same argument we also get that $f_A(\mathcal{C}_{i+1}) - f_A(\overline{\mathcal{C}}_{i+1}) \leq \epsilon/5$, which completes the proof. $\qquad\square$

From here our proof is somewhat similar to that of Section 3. Let us state the following useful lemma.

**Lemma 4.6.** *It holds w.h.p that for every $i \in [t]$ :*

$$f_X(\overline{\mathcal{C}}_{i+1}) - f_X(\mathcal{C}_{i+1}) \geq -\epsilon/5 \tag{1}$$

$$f_{B_i}(\mathcal{C}_{i+1}) - f_{B_i}(\overline{\mathcal{C}}_{i+1}) \geq -\epsilon/5 \tag{2}$$

$$f_X(\mathcal{C}_i) - f_{B_i}(\mathcal{C}_i) \geq -\epsilon/5 \tag{3}$$

$$f_{B_i}(\overline{\mathcal{C}}_{i+1}) - f_X(\overline{\mathcal{C}}_{i+1}) \geq -\epsilon/5 \tag{4}$$

*Proof.* The first two inequalities follow from Lemma 4.5. The last two are due to Lemma 3.1 by setting $\delta = \epsilon/5$, $B = B_i$:

$$Pr[|f_{B_i}(\mathcal{C}) - f_X(\mathcal{C})| \geq \delta] \leq 2e^{-2b\delta^2/d^2} = e^{-\Theta(b\epsilon^2/d^2)} = e^{-\Omega(\log(nt))} = O(1/nt)$$

Where the last inequality is due to the fact that $b = \Omega((d/\epsilon)^2 \log(nt))$ (for an appropriate constant). The above holds for either $\mathcal{C} = \mathcal{C}_i$ or $\mathcal{C} = \overline{\mathcal{C}}_{i+1}$. Taking a union bound over all $t$ iterations we get the desired result. $\qquad\square$

**Putting everything together**    We wish to lower bound $f_X(\mathcal{C}_i) - f_X(\mathcal{C}_{i+1})$. We write the following:

$$\begin{aligned}
f_X(\mathcal{C}_i) - f_X(\mathcal{C}_{i+1}) &= f_X(\mathcal{C}_i) \pm f_{B_i}(\mathcal{C}_i) - f_X(\mathcal{C}_{i+1}) \\
&\geq f_{B_i}(\mathcal{C}_i) - f_X(\mathcal{C}_{i+1}) - \epsilon/5 = f_{B_i}(\mathcal{C}_i) \pm f_{B_i}(\mathcal{C}_{i+1}) - f_X(\mathcal{C}_{i+1}) - \epsilon/5 \\
&\geq f_{B_i}(\mathcal{C}_{i+1}) - f_X(\mathcal{C}_{i+1}) + 4\epsilon/5 \\
&= f_{B_i}(\mathcal{C}_{i+1}) \pm f_{B_i}(\overline{\mathcal{C}}_{i+1}) \pm f_X(\overline{\mathcal{C}}_{i+1}) - f_X(\mathcal{C}_{i+1}) + 4\epsilon/5 \geq \epsilon/5
\end{aligned}$$

Where the first inequality is due to inequality (3) in Lemma 4.6 ($f_X(\mathcal{C}_i) - f_{B_i}(\mathcal{C}_i) \geq -\epsilon/5$), the second is due to the stopping condition of the algorithm ($f_{B_i}(\mathcal{C}_i) - f_{B_i}(\mathcal{C}_{i+1}) > \epsilon$), and the last is due to the remaining inequalities in Lemma 4.6. The above holds w.h.p over all of the iterations of the algorithms.

As in Section 3, we conclude that w.h.p it holds that $t = O(d/\epsilon)$, which implies that $b = \Omega((d/\epsilon)^2 \log(knd/\epsilon))$ is sufficient. We state our main theorem.

**Theorem 4.7.** *For $b = \tilde{\Omega}((d/\epsilon)^2)$ and $\alpha_i^j = \sqrt{b_i^j/b}$, Algorithm 1 with center update $\mathcal{C}_{i+1}^j = (1 - \alpha_i^j)\mathcal{C}_i^j + \alpha_i^j cm(B_i^j)$, terminates within $O(d/\epsilon)$ iterations w.h.p.*

## 5    Application to sklearn

In this section, we show the relevance of our results to the algorithm implementation of sklearn. The main differences in sklearn are the learning rate and stopping condition. The termination condition[4] depends on the movement of the centers in the iteration, rather than the value of $f_{B_i}$. Specifically, we continue as long as $\sum_{j \in [k]} \Delta(\mathcal{C}_{i+1}^j, \mathcal{C}_i^j) \geq \epsilon$ for some tolerance parameter $\epsilon$. The learning rate is set as $\alpha_i^j = \frac{b_i^j}{\sum_{\ell=1}^{i} b_\ell^j}$. Roughly speaking, this implies that $\alpha_i^j \to 0$ over time, and guarantees termination of the algorithm in the limit.

However, for our convergence guarantee, we only require $\alpha_i^j = \sqrt{b_i^j/b}$ which need not go to 0 over time. We show that with our learning rate and the termination condition of sklearn, the proof from Section 4 still implies termination, although at a slower rate and requires a larger batch size.

---

[4]The exact parameters of this algorithm were extracted directly from the code (the relevant function is _mini_batch_convergence): `https://github.com/scikit-learn/scikit-learn/blob/baf828ca1/sklearn/cluster/_kmeans.py#L1502`.

Specifically, we terminate within $O((d/\epsilon)^{1.5}\sqrt{k})$ iterations w.h.p if the batch size is $\tilde{\Omega}(k(d/\epsilon)^3)$. Note that this result is not subsumed by the result in Section 3 because the stopping condition is different.

Below we show that as long as the termination condition in sklearn does not hold ($\sum_{j\in[k]} \Delta(\mathcal{C}_{i+1}^j, \mathcal{C}_i^j) \geq \epsilon$), our stopping condition also does not hold for an appropriate parameter ($f_{B_i}(\mathcal{C}_i) - f_{B_i}(\mathcal{C}_{i+1}) > \epsilon'$ where $\epsilon' = \epsilon^{1.5}/\sqrt{kd}$). We state the following useful lemma:

**Lemma 5.1.** *Let* $x, y \in \mathbb{R}^d, \alpha \in [0,1]$. *It holds that* $\Delta(x, (1-\alpha)x + \alpha y) = \alpha^2 \Delta(x, y)$.

*Proof.* $\Delta(x, (1-\alpha)x + \alpha y) = \|x - (1-\alpha)x + \alpha y\|^2 = \|\alpha x - \alpha y\|^2 = \alpha^2 \Delta(x, y)$. $\qquad\square$

Below is our main lemma for this section:

**Lemma 5.2.** *If it holds that* $\sum_{j\in[k]} \Delta(\mathcal{C}_{i+1}^j, \mathcal{C}_i^j) > \epsilon$ *then* $f_{B_i}(\mathcal{C}_i) - f_{B_i}(\mathcal{C}_{i+1}) > \frac{\epsilon^{1.5}}{\sqrt{kd}}$.

*Proof.* Recall that $\mathcal{C}_{i+1}^j = (1-\alpha_i^j)\mathcal{C}_i^j + \alpha_i^j cm(B_i^j)$ for $\alpha_i^j = \sqrt{b_i^j/b}$. Thus, we get:

$$\epsilon < \sum_{j\in[k]} \Delta(\mathcal{C}_i^j, \mathcal{C}_{i+1}^j) \leq \sum_{j\in[k]} (\alpha_i^j)^2 \Delta(\mathcal{C}_i^j, cm(B_i^j)) = \sum_{j\in[k]} \frac{b_i^j}{b} \Delta(\mathcal{C}_i^j, cm(B_i^j)) \qquad (5)$$

Where in the transitions we used Lemma 5.1. Let us fix some $j \in [k]$, we can write the following:

$$\Delta(B_i^j, \mathcal{C}_i^j) - \Delta(B_i^j, \mathcal{C}_{i+1}^j)$$
$$= \Delta(B_i^j, cm(B_i^j)) + b_i^j \Delta(\mathcal{C}_i^j, cm(B_i^j)) - \Delta(B_i^j, cm(B_i^j)) - b_i^j \Delta(\mathcal{C}_{i+1}^j, cm(B_i^j))$$
$$= b_i^j (\Delta(\mathcal{C}_i^j, cm(B_i^j)) - \Delta(\mathcal{C}_{i+1}^j, cm(B_i^j)))$$
$$= b_i^j (\Delta(\mathcal{C}_i^j, cm(B_i^j)) - \Delta((1-\alpha_i^j)\mathcal{C}_i^j + \alpha_i^j cm(B_i^j), cm(B_i^j)))$$
$$= b_i^j (\Delta(\mathcal{C}_i^j, cm(B_i^j)) - (1-\alpha_i^j)^2 \Delta(\mathcal{C}_i^j, cm(B_i^j)))$$
$$= (2\alpha_i^j - (\alpha_i^j)^2) b_i^j \Delta(\mathcal{C}_i^j, cm(B_i^j)) \geq \alpha_i^j b_i^j \Delta(\mathcal{C}_i^j, cm(B_i^j)) =$$

Where in the first transition we apply Lemma 4.2, and in the last we use the fact that $\Delta(\mathcal{C}_{i+1}^j, cm(B_i^j)) = (\alpha_i^j)^2 \Delta(\mathcal{C}_i^j, cm(B_i^j))$ and the fact that $\forall, \alpha_i^j \in [0,1], 2\alpha_i^j - (\alpha_i^j)^2 \geq \alpha_i^j$. Let us bound $f_{B_i}(\mathcal{C}_i) - f_{B_i}(\mathcal{C}_{i+1})$:

$$f_{B_i}(\mathcal{C}_i) - f_{B_i}(\mathcal{C}_{i+1}) \geq \frac{1}{b} \sum_{j=1}^k (\Delta(B_i^j, \mathcal{C}_i^j) - \Delta(B_i^j, \mathcal{C}_{i+1}^j))$$

$$\geq \sum_{j=1}^k \frac{\alpha_i^j b_i^j}{b} \Delta(\mathcal{C}_i^j, cm(B_i^j)) = \sum_{j=1}^k \left(\frac{b_i^j}{b}\right)^{1.5} \Delta(\mathcal{C}_i^j, cm(B_i^j))$$

Where the first inequality is due to Observation 4.1, the second is due to the fact that $\forall j \in [k], \Delta(B_i^j, \mathcal{C}_i^j) - \Delta(B_i^j, \mathcal{C}_{i+1}^j) \geq \alpha_i^j b_i^j \Delta(\mathcal{C}_i^j, cm(B_i^j))$, and in the last equality we simply plug in $\alpha_i^j = \frac{b_i^j}{b}$ combined with. We complete the proof by applying Jensen's inequality, with parameters: $\phi(x) = x^{1.5}, y_j = b_i^j/b$ and $a_j = \Delta(\mathcal{C}_i^j, cm(B_i^j))$, combined with inequality (5).

$$\sum_{j=1}^k \left(\frac{b_i^j}{b}\right)^{1.5} \Delta(\mathcal{C}_i^j, cm(B_i^j)) \geq \left(\sum_{j=1}^k \Delta(\mathcal{C}_i^j, cm(B_i^j))\right) \cdot \left(\frac{\sum_{j=1}^k \frac{b_i^j}{b}\Delta(\mathcal{C}_i^j, cm(B_i^j))}{\sum_{j=1}^k \Delta(\mathcal{C}_i^j, cm(B_i^j))}\right)^{1.5}$$

$$\geq \frac{\epsilon^{1.5}}{\sqrt{\sum_{j=1}^k \Delta(\mathcal{C}_i^j, cm(B_i^j))}} \geq \frac{\epsilon^{1.5}}{\sqrt{kd}}$$

$\qquad\square$

Finally, plugging $\epsilon' = \frac{\epsilon^{1.5}}{\sqrt{kd}}$ into our bounds, we conclude that if $b = \tilde{\Omega}(\epsilon^{-3}d^3k)$ then the number of iterations is bounded by $O((d/\epsilon)^{1.5}\sqrt{k})$ w.h.p.

ACKNOWLEDGMENTS

The author would like to thank Ami Paz, Uri Meir and Giovanni Viglietta for reading preliminary versions of this work.

This work was supported by JSPS KAKENHI Grant Numbers JP21H05850, JP21K17703, JP21KK0204.

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
