# OpenReview forum: "Mini-batch $k$-means terminates within $O(d/\epsilon)$ iterations"
_ICLR.cc/2023/Conference — ICLR 2023 poster_

### Official Review · Reviewer_VXtC · 2022-10-23

**Confidence:** 3
**Correctness:** 3
**Technical Novelty And Significance:** 2
**Empirical Novelty And Significance:** Not applicable
**Recommendation:** 6

**Clarity, Quality, Novelty And Reproducibility:**

# Clarity:

Some places are not written in a rigorous way. Detailed comments:

1. Page 3, the meaning of the notation \Delta(x, C) is not defined; in fact, it is defined that \Delta(S, x) being the sum of \Detal(y, x) for y \in S, but this \Delta(x, C) actually means a different thing. Please consider to improve the notation.
2. In Theorem 2.2, it seems \delta is not quantified. In the summation inside \Pr, the index is I but you used Y_k.
3. I find the definition of learning rate in Sec 1 quite confusing. Also, I don’t find a place that this learning rate is formally defined.
4. It is suggested to add formal theorem statements for your main results.

Originality:

This paper studies a widely used clustering method that is currently not fully understood, hence is timely. However, the techniques are mostly based on standard concentration bounds, which are not particularly novel. At first glance, the main result seemed to be surprising since the number of samples as well as the number of iterations are independent of both k and n which are the fundamental parameters in clustering, but as mentioned in the “Weakness”, this bound is of a cost of accuracy.


**Strength And Weaknesses:**

# Strength:

Mini-batch clustering is a practical k-means implementation, hence it is well motivated to analyze the performance of the algorithm. An analysis/comparison to the implementation in sklearn is also welcome.

# Weakness:

An empirical evaluation that validates the number of iterations and the accuracy is very interesting. Unfortunately, no empirical evaluation is provided at all.

Another perhaps more severe limitation is that only the number of iterations is analyzed, and the accuracy is not discussed. In particular, I don’t think sampling o(k) points (which the authors do) can lead to any finite approximation. For instance, consider 1D line, and there’re only k distinct points, but they each have multiplicities of e.g., 100. Then, a uniform sample of o(k) points cannot discover all of these k distinct points, and eventually the output center set must also contain less than k distinct points. However, the optimal solution value is 0 which is achieved by simply putting one center point at each distinct point, while what you obtain must have a cost > 0, hence the multiplicative error is unbounded.


**Summary Of The Paper:**

This paper analyzes the mini-batch k-means algorithm, a version of which is also used in the well-known sklearn. The mini-batch k-means is modeled in the following generic way: in every iteration, sample b (for some b which they bound) uniform samples from the entire dataset X, and run the Lloyd step only on the sampled set.

The main result shows that if b = \tilde{\Omega}(d^2 \eps^{-2}), then the entire algorithm terminates in O(d / eps) iterations, w.h.p., where eps is the termination threshold. An analysis/comparison to the sklearn’s implementation is also provided.

**Summary Of The Review:**

Even though the techniques are somewhat standard, the obtained bounds look nice. However, I'd still like to reject the paper, since I believe the paper misses an important aspect, which is the accuracy of the algorithm (as discussed in "Weakness"), and that I don't see the empirical evaluation that justifies the accuracy.

---

> ### Author Response · Authors · 2022-11-11
> **Authors' response to Reviewer VXtC**
>
> Thank you very much for taking the time to read this paper. Please find our detailed response below.
>
> >Another perhaps more severe limitation is that only the number of iterations is analyzed, and the accuracy is not discussed. In particular, I don’t think sampling o(k) points (which the authors do) can lead to any finite approximation. For instance, consider 1D line, and there’re only k distinct points, but they each have multiplicities of e.g., 100. Then, a uniform sample of o(k) points cannot discover all of these k distinct points, and eventually the output center set must also contain less than k distinct points. However, the optimal solution value is 0 which is achieved by simply putting one center point at each distinct point, while what you obtain must have a cost > 0, hence the multiplicative error is unbounded.
>
> Indeed the quality of the solution is not discussed in this paper. Please see our response to Reviewer 8vVB for a detailed discussion. Nevertheless, our algorithms with k-means++ initialization achieve the same approximation guarantees. For your example, the k-means++ initialization procedure will select the k unique centers. If our batch is o(k) indeed not all clusters will have a point selected. This will result in some clusters not having any points assigned to them during the iteration. Note that we do not remove clusters when no points in the batch are assigned to them. So for your example, the clusters will not move at all, the algorithm will terminate, and we will get the optimal solution.
>
> Please also note that applying our bounds to the sklearn implementation introduces a linear dependence in k in the batch size, which circumvents this issue.

---

> > ### Comment · Reviewer_VXtC · 2022-12-07
> > **Followup questions**
> >
> > Thanks for the response. Indeed, one needs to use a certain initialization to avoid the instance that I mentioned.
> >
> > However, I'm still not sure whether your algorithm, especially the specific version mentioned in Sec 4, preserves the ratio of the initialization in general. Here, I'm thinking about a statement like this: given an \alpha-approximate initialization, the algorithm ends up with an O(\alpha)-approximate solution. I saw you mentioned something similar in the response to reviewer 8vVB where you claimed your update rule must improve the quality of the solution, but this is not immediate to me. In particular, your update rule is C_{i + 1} = (1 - \alpha) C_i + \alpha cm(S), where S is a uniformly-random sample, so clearly it is possible that the quality of the solution can be worse after one update (due to the randomness). Hence, we need an argument for the overall accuracy guarantee of your algorithm, which may be true but seems nontrivial to me.

---

> > > ### Comment · Reviewer_VXtC · 2022-12-07
> > > **-**
> > >
> > > After a closer look at the paper, it seems the authors already show the algorithm improves the cost in every iteration w.h.p. (in the paragraph above Theorem 4.7). So the algorithm indeed preserves the ratio of the initialization, which resolves my concern/question.
> > >
> > > I'd like to raise my score, and the authors should add the relevant discussion with 8vVB about the accuracy (possibly with more details) to the next version.

---

> > > > ### Author Response · Authors · 2022-12-07
> > > > **Thank you**
> > > >
> > > > Thank you very much! We will make sure to add the relevant discussion to the final version of the paper.

---

### Official Review · Reviewer_yi8Y · 2022-10-31

**Confidence:** 3
**Correctness:** 4
**Technical Novelty And Significance:** 2
**Empirical Novelty And Significance:** Not applicable
**Recommendation:** 5

**Clarity, Quality, Novelty And Reproducibility:**

The paper is clearly written, barring  some typos. It is reproducible as mostly it consists of theoretical analysis. The analysis is moderately novel.

**Strength And Weaknesses:**

Strengths: The strength of the paper is that it investigates an important problem and establishes a nice upper bound result. The analysis is also nice and simple.

Weakness: The research question is very interesting but is in the initial stages. The one result established is nice (and the reviewer is pleasantly surprised that it is not considered earlier) but the picture is far from complete. In particular, the lower bound landscape is completely missing. It will be nice to prove that for $O(d/\epsilon)$ convergence, the sample complexity (or the mini batch size) has to be $\Omega(d^2/\epsilon^2)$. If not why the present algorithm is not optimal? Any concert result indicating that current bounds are optimal will be nice.  More generally is there a generic trade off between the convergence rate and the batch size? Also, as the authors point out, any guarantee on the the quality of solution (does it converge to a local minima) is missing.

**Summary Of The Paper:**

This paper investigates  the convergenence rate of the mini-batch $k$-means algorithm. In the well-known k-means clustering problem, we are given a set of $n$ points and a number $k$ and the goal is to find $k$ centers (the center set) that minimizes the sum (or average as in this paper) of the squared distances of all the points to the nearest center point. The well-known Lloyd's algorithm starts with a random $k$ centers, partitions the $n$-point set into $k$ sets by assigning each point to its nearest center and then computes the centriods of the partition as the new center set and repeats (until a stopping criteria is met).

In the mini-batch version, which is analyzed in this paper, instead of considering all the $n$ points in each step to update the centers, the algorithm independently and randomly picks $b$ (batch size) points from the point set and performs the center update. This reduces the running time of each iteration. The question that is investigated is how fast this mini-batch algorithm converges: For the given set of $n$ points and a center set $C$, let $f(C)$ be the quantity (average of the squared distances) that we would like to optimize. Then the algorithm at iteration $i$, computes the center set $C_i$. We say that the algorithm $(t,\epsilon)$-converges if $f(C_t)-f(C_{t+1}) < \epsilon$. Clearly if the diameter of the universe is bounded by $d$, then the original algorithm converges in $d/\epsilon$ steps ($f(C_{I+1} \leq f(C_i)$ at all $i$ since the centroid minimizes the sum of squared distances). The paper considers the problem: what is the batch size $b$ that is required to achieve asymptotically the same convergence rate. The paper establishes that if  $b =  \tilde\Omega(d^2/\epsilon^2)$, a variant of the mini-batch k-means algorithm  $(O(d/\epsilon),\epsilon)$ converges.


**Summary Of The Review:**

The main result is nice and publishable. The main criticism is the completeness of the research conducted. There are many missing issues (lower bounds and tradeoffs) which if addressed will make this submission into a solid paper.

---

> ### Author Response · Authors · 2022-11-11
> **Authors' response to Reviewer yi8Y**
>
> Thank you very much for taking the time to read this paper. Please find our detailed response below.
>
> >The research question is very interesting but is in the initial stages. The one result established is nice (and the reviewer is pleasantly surprised that it is not considered earlier) but the picture is far from complete. In particular, the lower bound landscape is completely missing. It will be nice to prove that for  convergence, the sample complexity (or the mini batch size) has to be . If not why the present algorithm is not optimal? Any concert result indicating that current bounds are optimal will be nice. More generally is there a generic trade off between the convergence rate and the batch size? Also, as the authors point out, any guarantee on the the quality of solution (does it converge to a local minima) is missing.
>
> We agree that adding lower bounds to our results will strengthen the paper. We carefully considered this direction and we believe that a lower bound of $b=\Omega(d/\epsilon)$ can be proved. However, properly writing this bound such that it holds for all values of $d,k$ might be hard to achieve within the revision phase. Therefore, we would like to defer this to future work.
>
> As for the quality of the solution, please see our response to reviewer 8vVB.

---

### Official Review · Reviewer_8vVB · 2022-11-02

**Confidence:** 3
**Correctness:** 4
**Technical Novelty And Significance:** 3
**Empirical Novelty And Significance:** Not applicable
**Recommendation:** 6

**Clarity, Quality, Novelty And Reproducibility:**

Quality:
- the paper studies thoroughly from a theoretical perspective the convergence of minibatch k-means. It would have been nice to see some connections to the real-world usage of k-means.

Clarity:
- the paper is well-written, and the steps followed by the authors to prove their theorems are intuitive and natural.
- the authors are concise, and only included the information needed for the paper. This has led to a short paper that discusses well the story, without the need for additional jargon of text. (positive feedback)

Originality:
- The authors are aware of one other related to work that analyzes the convergence of the minibatch k-means. The other work depends on the input size, while this work does not.

**Strength And Weaknesses:**

Strength:
- well-written paper and easy to follow
- the transferability of the results to a popular library (sklearn) with minor modifications
- the paper shows that under some conditions, a progress on one batch of k-means leads to global progress, hence supporting the claim that minibatch k-means could be used when computation is a bottleneck

Weaknesses:
- the limiting bounds are nice theoretically, however, from a practical perspective, it would be nice to get a perspective of what these numbers would be. Specifically, the implementation of minibatch k-means in sklearn already has good hyperparameters to start with, and the results are usually good without any tweak. What would be for example the required batch size in order to terminate as fast as the default number of iterations?
- as pointed out by the authors, the paper does not address the quality of the solution, but rather focuses on runtime. From a practical perspective, the quality of the solution is of essence. The obtained solution does not need to amazing, but the quality of the solution should not degrade much. It would be nice to get some analysis about this aspect.

**Summary Of The Paper:**

The paper studies from a theoretical perspective the convergence of mini-batch k-means. Traditionally, approaches to mini-batch k-means execute the algorithm for a fixed number of iteration, or until a convergence criterion is met (early stopping). The authors of this paper consider the latter case, i.e. mini-batch k-means algorithms without a fixed number of steps. Within this setup, the authors identify conditions under which the algorithm terminates with high probability rather than run forever. The conditions outlined in the paper relate to the batch size used, and are independent of the size of the dataset, or the initialization of the clusters.

**Summary Of The Review:**

The paper presents support theoretically the claim that local progress on minibatch k-means leads to progress on the global objective, assuming the batch size is large enough. This goes in hand with the proof that the algorithm terminates under some conditions without the need for setting a fixed number of iterations. Although the result is nice theoretically, adding some analysis about practical implications would be a big plus, e.g. what would be an estimate of the batch size for some scenarios, and empirically verifying the claims. Furthermore, discussing slightly the quality of the solution would be good in order to have a better idea about the tradeoffs.

---

> ### Author Response · Authors · 2022-11-11
> **Authors' response to Reviewer 8vVB**
>
> Thank you very much for taking the time to read this paper. Please find our detailed response below.
>
> >the limiting bounds are nice theoretically, however, from a practical perspective, it would be nice to get a perspective of what these numbers would be. Specifically, the implementation of minibatch k-means in sklearn already has good hyperparameters to start with, and the results are usually good without any tweak. What would be for example the required batch size in order to terminate as fast as the default number of iterations?
>
> This is somewhat difficult to answer as this would depend on the dimension of the data ($d$) and the termination sensitivity ($\epsilon$). Currently, sklearn does not provide a default value for the termination sensitivity and uses a different early stopping criteria as the default one.
>
> > as pointed out by the authors, the paper does not address the quality of the solution, but rather focuses on runtime. From a practical perspective, the quality of the solution is of essence. The obtained solution does not need to amazing, but the quality of the solution should not degrade much. It would be nice to get some analysis about this aspect.
>
> When talking about quality, there are two natural ways to define it: (1) Good approximation ratio (e.g., k-means++) (2) Local minima of the goal function (e.g., Lloyd's algorithm). Let us address both of these points.
>
> 1) Our algorithm initialized with the k-means++ initialization achieves the same approximation ratio, $O(\log k)$ in expectation. The approximation guarantee in k-means++ is guaranteed already in the initialization phase (Theorem 3.1 in https://theory.stanford.edu/~sergei/papers/kMeansPP-soda.pdf), and the execution of Lloyd's algorithm following initialization can only improve the solution. We also show that, w.h.p, the global goal function is decreasing throughout our execution. Therefore, our center updates also only improve the quality of the solution, and the approximation guarantee remains the same.
>
> 2) As long as the local progress condition holds, we achieve an additive improvement in the goal function over the entire dataset. Therefore the longer the algorithm executes, the closer it is to a global optimum. However, due to the local progress condition, the algorithm may terminate prematurely, so we do not think that it is possible to guarantee the result is close to a local minimum (i.e., an output of Lloyd's algorithm) without making additional assumptions on the input (e.g., clusterability). Adding the local progress assumption to Lloyd's algorithm (non mini-batch) already breaks its guarantee of reaching a local minimum. Showing convergence guarantees (perhaps under some assumptions), is indeed an interesting and natural research direction. Nevertheless, we still believe that even termination guarantees are very interesting for such a widely used algorithm.

---

### Official Review · Reviewer_hFMG · 2022-11-07

**Confidence:** 5
**Clarity, Quality, Novelty And Reproducibility:** The paper is clearly written and easy…
**Correctness:** 4
**Technical Novelty And Significance:** 4
**Empirical Novelty And Significance:** Not applicable
**Recommendation:** 10

**Strength And Weaknesses:**

This is an appealing result, the best in my small pile, and should definitely be accepted.

**Summary Of The Paper:**

The paper analyzes the convergence rate of mini-batch k-means, namely, running Lloyd's iteration with a uniform sample of points from the data set, rather than using the entire set in each iteration. It gives strong results: with a sample size nearly quadratic in the dimension, the number of steps needed is linear in the dimension (and independent of the size of the data set). This requires a stopping condition that deviates from practice, and somewhat weaker bounds are shown for the conventional stopping condition.

**Summary Of The Review:**

The convergence bound is strong, and the paper actually indicates a modification in the standard implementation that could result is superior performance in practice.

---

> ### Author Response · Authors · 2022-11-11
> **Authors' response to Reviewer hFMG**
>
> Thank you very much for taking the time to read this paper. As an author, it is truly a pleasure to read such a positive review.

---

### Decision · Program_Chairs · 2023-01-20

**Decision:**

Accept: poster

**Justification For Why Not Higher Score:**

Good paper, could be pushed up a bit.

**Justification For Why Not Lower Score:**

Clearly above acceptance threshold.

**Metareview: Summary, Strengths And Weaknesses:**

This paper shows that mini-batch k-means terminates as long as the batch sizes are sufficiently large, with high probability. The result is neat and makes progress on an important algorithm that is also used in practice.

**Note From Pc:**

if the above contains the word "oral" or "spotlight" please see: "oral" presentation means -> notable-top-5% and "spotlight" means -> notable-top-25%. As stated in our emails, we are disassociating presentation type from AC recommendations

**Summary Of Ac-Reviewer Meeting:**

There was a combination of email discussion and video discussion for this paper. Due to timezone issues separate meetings had to be carried out. After the meetings and further email discussion, it was agreed that the paper should be accepted.